# Control of the Organization of 4,4′-bis(carbazole)-1,1′-biphenyl (CBP) Molecular Materials through Siloxane Functionalization

**DOI:** 10.3390/molecules28052038

**Published:** 2023-02-21

**Authors:** Janah Shaya, Jean-Charles Ribierre, Gabriel Correia, Yannick J. Dappe, Fabrice Mathevet, Loïc Mager, Benoît Heinrich, Stéphane Méry

**Affiliations:** 1Institut de Physique et Chimie des Matériaux de Strasbourg (IPCMS), CNRS, Strasbourg University, UMR 7504, 23 rue du Loess, 67034 Strasbourg, France; 2Department of Chemistry, College of Arts and Sciences, Khalifa University, Abu Dhabi P.O. Box 127788, United Arab Emirates; 3College of Medicine and Health Sciences, Khalifa University, Abu Dhabi P.O. Box 127788, United Arab Emirates; 4Service de Physique de l’État Condensé, CEA CNRS UMR 3680, Université Paris Saclay, 91191 Gif-sur-Yvette, France; 5Institut Parisien de Chimie Moléculaire (IPCM), CNRS, Sorbonne University, 4 Place Jussieu, 75005 Paris, France; 6Center for Organic Photonics and Electronics Research (OPERA), Department of Applied Chemistry, Kyushu Universty, 744 Motooka, Nishi, Fukuoka 819-0395, Japan

**Keywords:** molecular liquid, liquid crystal, siloxane side-chain, siloxane-terminated side chain, hybrid siloxane-based chain, liquid electronics, liquid emitter

## Abstract

We show that through the introduction of short dimethylsiloxane chains, it was possible to suppress the crystalline state of CBP in favor of various types of organization, transitioning from a soft crystal to a fluid liquid crystal mesophase, then to a liquid state. Characterized by X-ray scattering, all organizations reveal a similar layered configuration in which layers of edge-on lying CBP cores alternate with siloxane. The difference between all CBP organizations essentially lay on the regularity of the molecular packing that modulates the interactions of neighboring conjugated cores. As a result, the materials show quite different thin film absorption and emission properties, which could be correlated to the features of the chemical architectures and the molecular organizations.

## 1. Introduction

The control of molecular organization is the object of constant and intense research activity since it determines most properties of solid-state materials. So far, a considerable amount of work has been published to describe the role of molecular parameters acting on various molecular forces and on steric effects, in particular, to try to control the solid-state molecular packing [1,2,3,4]. As an example, the substitution of organic conjugated molecules by flexible peripheral chains has become a convenient tool to control, to some extent, the molecular organization via the tuning of van der Waals interactions and steric constraints [5,6,7]. Until now, most of studies on side-chain functionalization have focused on alkyl chains. This is due to the wide variety of commercially available alkyl chains, including ramified ones. In comparison, very few studies have been performed with other types of chains, such as ethyleneoxide [8] or siloxane chains [9], which may induce different organization properties due to their distinctive features.

Thus, oligodimethylsiloxane (ODMS) chains -(SiMe_2_-O)_x_- are primarily characterized by exceptional flexibility which accounts for most of their unique properties [10]. This peculiar feature arises from the nearly free torsional motion along the Si-O backbone [11]. Hence, ODMS hardly crystallizes but exhibits a strong amorphous character, as illustrated by the very low glass transition of its parent polydimethylsiloxane (PDMS) (*T_g_* = −120 °C) [12]. Siloxane backbone also contains weak dipole moments, but the latter can easily be masked or uncovered (by an easy umbrella-type motion of the pending methyl groups), enabling ODMS to readily adapt to a polar or apolar environment [11]. Overall, ODMS possesses weak intermolecular forces entailing a very low surface tension, solubility parameter and dielectric constant, in particular [13]. In addition, ODMS is endowed with a propensity for microphase separation easily leading to the self-organization of siloxane-containing molecular systems [14,15,16]. Finally, with regard to geometrical parameters, siloxane chains are rather bulky, as illustrated by the larger molecular section of the linear ODMS chain (*σ* = 41 Å^2^) as compared to the one of the linear alkyl chains (*σ* = 21.5 Å^2^) [17].

The specific features of siloxanes have stimulated the development of many siloxane-containing organic compounds. Thus, siloxane functionalization has been applied to control the organization and tune properties of many molecular and macromolecular systems for a wide range of application domains, including optoelectronics [9,18,19]. For example, functionalization with ODMS has been used to obtain liquid crystalline organizations with favorable *π*-molecular interactions for charge transport properties [20]. A complex donor-acceptor (D-A) nanostructured smectic phase could even be stabilized by incorporating ODMS segments at both extremities of a D-A-D molecular triad [21,22]. A number of siloxane-hybrid side-chains conjugated polymers have also been reported to exhibit lamellar mesomorphic organization with enhanced charge transport properties (e.g., with mobilities around 1 cm^2^ V^−1^ s^−1^) [23,24,25]. This effect was recently attributed to the fluid and nanosegregating character of siloxane chains that impose a better facing of the polymer backbones with improved *π*-stacking overlap [17]. In other respects, the bulky and flexible character of ODMS has also been used to deliberately hinder *π*-intermolecular interactions to obtain solvent-free *π*-functional molecular liquids at room temperature. Thus, multiple siloxane functionalization of different arylamine and fluorene derivatives led to room-temperature liquid materials with significant charge transport and emission properties in their neat liquid phase [26,27], anticipating promising future applications of siloxane-functionalized materials in liquid (opto)electronics [28,29,30].

The study presented herein aims at using siloxane functionalization for tuning the molecular organization of a representative *π*-conjugated molecule used in optoelectronics. The objective was to scrutinize how the variation in the location and proportion of siloxane chains is able to modify and control the molecular organization from the initial system. For this study, we selected 4,4’-bis(carbazol-9-yl)biphenyl (also called CBP), a well-known carbazole-based material used as a host in organic light emitting devices [31,32,33]. CBP crystallizes in a herringbone-type packing and exhibits a high melting point (282 °C) [34,35]. In devices, CBP is used in its metastable glassy state obtained by chilling, but it strongly suffers from its tendency to return to its natural crystalline state [35]. Thus, rendering CBP a molecular liquid or a liquid crystal through siloxane functionalization constitutes a route toward morphologically stable guest-host devices and/or fluidic devices [29,30,36,37].

The synthetic procedure to prepare the siloxane-functionalized CBP has recently been reported elsewhere [38]. The latter focusses on the methodology we followed to introduce one or two short heptamethylsiloxane segments (via a propylene linker) at each of the carbazole end-units of CBP, to yield CBP-2Si_3_ and CBP-4Si_3_, respectively (Figure 1). As a preliminary result, CBP-2Si_3_ showed a considerable drop of the melting point (T_m_ = 87 °C) as compared to native CBP, and CBP-4Si_3_ exhibited a stable and fluid liquid state at room temperature, for which the only thermal event observed was a glass transition at *T_g_* = −62 °C. These first observations already point to the considerable impact of the siloxane functionalization on the molecular organization of CBP derivatives. The aim of the present study is to undertake an extensive structural characterization by means of X-ray scattering to unravel the role of the siloxane chains on the fine molecular organization of the siloxane-functionalized CBP derivatives. In this paper, we will show in particular that the minimal insertion of siloxane in CBP-2Si_3_ is able to stabilize a lamellar organization which can evolve up to the formation of a fluid liquid crystalline smectic phase after lengthening the siloxane chain in CBP-2Si_n_ (with *n* ≈ 10), a new compound reported herein (Figure 1). Finally, the liquid state of CBP-4Si_3_ reveals the presence of structuration at the local range that will be detailed and discussed. All these changes in molecular organization naturally impact the neat film absorption and emission properties that will also be analyzed herein, in relation to the fine molecular packing.

## 2. Results and Discussion

The siloxane-based CBP derivatives investigated in this study are presented in Figure 1. Molecules CBP-2Si_3_ and CBP-4Si_3_ are substituted by two and four short heptamethyltrisiloxane chains, respectively, while CBP-2Si_n_ is substituted by 2 longer oligo(dimethylsiloxane) chains (polydispersity *Đ* = 1.2, with an average number of dimethylsiloxane units of 10, see Appendix A).

### 2.1. Synthesis

The different siloxane-functionalized CBP derivatives investigated in this study were synthesized using optimized catalytic methodologies recently reported elsewhere (Figure 1) [38]. Briefly, monobromocarbazole **1** was used to prepare the two CBP derivatives functionalized with two siloxane segments (short: CBP-2Si_3_ and longer: CBP-2Si_n_). A propylene linker was introduced to **1** via Stille cross-coupling leading to intermediate **3**. Then, **3** was reacted under Ullmann coupling conditions with dibromobiphenyl **5** to generate adduct **6**, which was further hydrosilylated with siloxane chains of different lengths to produce the final molecules CBP-2Si_3_ and CBP-2Si_n_. CBP-4Si_3_ derivative was prepared by the same sequence of reactions using 3,6-dibromocarbazole **2** instead of monobromocarbazole. The latter synthetic route involved to double the catalytic loadings and stoichiometry of reagents in the first and third steps, leading to intermediates **4**, **7**, and the final CBP derivative (CBP-4Si_3_) functionalized with 4 short siloxane segments instead of 2. The detailed synthetic protocols can be found in Ref. [38] or in the Appendix A.

### 2.2. Thermal and Structural Properties

The functionalization of the CBP core by siloxane chains is found to drastically impact the organization of the materials, as CBP-2Si_3_, CBP-2Si_n_, and CBP-4Si_3_ exhibit at room temperature a solid state, a liquid crystalline phase, and a liquid state, respectively. Table 1 summarizes the transition temperatures and the enthalpy changes, issued from differential calorimetry (DSC) and thermogravimetry (TGA) analyses, shown in Figure 2 and Appendix A, respectively.

To start with thermal stability, the siloxane chain functionalization lowers the materials stability as *T*_deg_ is decreased by about 100 °C for all siloxane-based CBP derivatives as regards to reference CBP. Next, we examine the effect of siloxane chain substitution on the organization of the CBP materials (See Table 1). The introduction of two short siloxane chains is sufficient to strongly destabilize the crystal state, as *T*_m_ is decreased from 282 to 87 °C when transitioning from CBP to CBP-2Si_3_. The destabilization is further aggravated by lengthening the siloxane chain, the crystalline state being replaced by a room-temperature liquid crystalline phase (clearing temperature: *T*_LC→Iso_ = 27 °C) in CBP-2Si_n_. Finally, the substitution of four short siloxane chains totally suppresses any long-range order in the packing of the CBP core, leading to a room-temperature liquid state with a sub-ambient glass transition temperature (*T*_g_ ≈ −60 °C) in CBP-4Si_3_. The destabilizing effect of the siloxane functionalization is well illustrated by the strong decrease of the enthalpy change, Δ*H*_→Iso_ (associated with the transition toward this isotropic liquid phase), observed in the CBP materials series. This value is found to drop stepwise when transitioning from the crystal CBP (40 kJ mol^−1^), the solid state CBP-2Si_3_ (16 kJ mol^−1^), the liquid crystalline CBP-2Si_n_ (11 kJ mol^−1^), down to the glassy CBP-4Si_3_ (0 kJ mol^−1^). The states of the CBP derivatives have been primarily assigned after polarized microscopic observation (POM). For CBP-2Si_n_, the liquid crystal phase is clearly evidenced by the POM texture obtained under crossed polarizers shown in Figure 3. The fluid focal-conic domain texture observed gives indication of the presence of a uniaxial lamellar mesophase.

The fine structural organization of the siloxane-functionalized CBP derivatives has been extensively investigated by small- and wide-angle X-ray scattering (SWAXS) and grazing incidence wide-angle X-ray scattering (GIWAXS). Representative SWAXS patterns recorded for all materials at room temperature are presented in Figure 4a–d. The GIWAXS patterns specifically recorded for CBP-2Si_3_ are shown in Figure 5 (room-temperature) and Appendix A (100 °C), respectively. A compilation of the structural parameter data can be found in Appendix A.

Examining the SWAXS pattern of CBP-2Si_3_ (at 20 °C), shown in Figure 4a, this pattern displays many sharp reflections in the whole angular range together with a broad scattering signal from lateral interactions of molten alkyl chains (*h*_ch_). This information indicates a soft-crystalline organization, in which crystal-like, three-dimensional long-range ordering of conjugated segments coexists with molten chain zones [39,40,41]. Structural features could be specified through the combination with an oriented GIWAXS pattern obtained for a spin-coated thin film of CBP-2Si_3_ (see Figure 5). Thereby, the structure and morphology of the film was lamellar with the layers oriented parallel to the substrate. The direction of the normal layer was identified to the *c*-axis of an orthorhombic structure, designed by the in-plane arrangement and the superposition of layers. The lattice parameters are reported in Appendix A. The cell periodicity involves two lamellae with a staggered superposition causing the extinction of (00*l*) reflections with odd *l* value; the lamellar periodicity is hence: *d*_lam_ = *d*_002_ = 17.25 Å. The in-plane arrangement follows an **a** × **b** sublattice of *p*2*mg* or *p*2*gg* symmetry that involves *Z*_2D_ = *Z*/2 = 2 molecular stacks covering an area *A*/*Z*_2D_ = *ab*/2 = 91.0 Å^2^ (cf. Appendix A). In addition to that of odd (00*l*), one notices the extinction of reflections (*h*00) and (*h*0*l*) with odd *h* value. Three space groups are then compatible with this composition of patterns: *Pca*2_1_, *Pna*2_1_, and *Pbcm*.

Figure 4b presents the SWAXS pattern of the liquid CBP-4Si_3_ recorded at 20 °C. A similar pattern was obtained for CBP-2Si_3_ in its high temperature (100 °C) liquid state (see Appendix A). Both patterns show the same distinct scattering signals for siloxane segments (*h*_dMS_ = 6.5–7 Å), and aliphatic or CBP segments (*h*_ch_ + *h*_ar_ = 4.5–5 Å), demonstrating the persistence of nanosegregated strata in the liquid state. The periodicity of the strata alternation leads to a further scattering signal in the low-angle region (*D*_lay_ = 29 and 27 Å, for CBP-2Si_3_ and CBP-4Si_3_, respectively, with similar correlation length *ξ* ≈ 60 Å obtained from full width half-maximum *FWHM* and Scherrer formula *ξ = K 2π/*Δ*q* with Δ*q* = (*FWHM*^2^—*FWHM*_0_^2^)^0.5^, beam width *FWHM*_0_ = 0.006 Å^−1^ and shape factor *K* = 0.9).

Figure 4c,d show the SWAXS patterns of CBP-2Si_n_ after different thermal treatments. They correspond to a record of the material at 10 °C in the mesophase (Figure 4c), and after heating to isotropic liquid down to the supercooled isotropic liquid at 20 °C (Figure 4d). First, the mesophase pattern (Figure 4c) indicates a mesophase with a smectic E-analogue structure, with lamellae constituted by the alternation of molten chain layers and layers of mesogens arranged in a long-range correlated two-dimensional rectangular lattice (lamellar periodicity: *d*_lam_ = *d*_001_ = 48.1 Å). The presence of the molten chain sublayers is indeed demonstrated by their characteristic signatures *h*_dMS_ and *h*_ch_. The lattice geometry and reflection intensity ratios of CBP-2Si_n_ are comparable to the *p*2*mg*
**a** × **b** sub-lattice of the soft crystal phase of CBP-2Si_3_, indicating the similarity of the molecular organizations, notwithstanding the loss of the three-dimensional superstructure as a result of the thick siloxane layer intercalation. Due to its low clearing temperature (24 °C), CBP-2Si_n_ stays for days in the isotropic liquid phase at room-temperature once melted. Then, the SWAXS pattern recorded in its supercooled liquid state at 20 °C (see Figure 4d) is very similar to the one of CBP-2Si_3_ and CBP-4Si_3_ in their isotropic liquid phase, as shown in Appendix A and Figure 4b, respectively. Because of the high proportion of siloxane chains in CBP-2Si_n_, its liquid state gives higher layer periodicity (*D*_lay_ = 42 Å) and correlation length (*ξ* ≈ 100 Å), as CBP-2Si_3_ and CBP-4Si_3_ in their liquid phase (compare with *D*_lay_ = 27–29 Å, and *ξ* ≈ 60 Å).

The combination of the results obtained by SWAXS measurements and by geometrical calculations gave access to a number of structural parameters (see Appendix A) allowing us to fully describe the molecular organization of the CBP derivatives in the different phases. First, let us consider the molecular organization of CBP-2Si_3_ in its soft crystal phase. Actually, the CBP core has a shape of a dumbbell of approximately 19 Å length and 9 Å width (both including the van der Waals radii), with lateral close-packing distances in the order of 4 Å, while siloxane units can be approximated by cylinders of 7–7.5 Å diameter. The nanosegregation of both moieties (i.e., aromatic core and siloxane chains) into lamellae implies that the molecular area of the sequence of layers fits the individual space requirements, which is realized by the interdigitation of the siloxane end-segments. The different types of self-arrangements (namely, rows aligned on the *a*-axis and spaced by *b*/2 = 4.3 Å for CBP, and close-packed cylinders for siloxane chains), are however mutually constrained by the interconnecting propylene spacer. Therefore, the successive rows constituting the CBP layers are longitudinally shifted along the *a*-axis to allow the close-packing of siloxane chains, which determines the in-plane periodicity of two rows along the *b*-axis. Additionally, the interdigitation of the side-chains imposes a staggered superposition of successive CBP layers and thus the periodicity of two molecular layers along the *c*-axis. These constraints result in an original *p2mg* arrangement of CBP cores and a cohesive three-dimensional structure of *Pca*2_1_ symmetry (Figure 6), which is consistent with the selected space groups.

The organization of CBP derivatives evolves significantly with siloxane chain content. The molecular self-organization can thus be controlled through the siloxane chain functionalization, as illustrated in Figure 7. Due to the lower cross-sectional area of the siloxane chain (about two-fold smaller) as compared to the CBP core, the two trisiloxane chains in CBP-2Si_3_ form intercalated monolayers that alternate with CBP monolayers. This intercalation strongly constrains the respective positions of segments from different layers, which explains the evolution of the lamellar structure into a cohesive three-dimensional soft-crystal. Conversely, these constraints are removed with the four chains of CBP-4Si_3_ and the disentanglement of the siloxane segments, resulting in a single liquid phase. On the other hand, the use of longer siloxane chains for CBP-2Si_n_ blurs the three-dimensional structure interconnecting CBP segments, while reinforcing the nanosegregation into layers. This results in the substitution of the lamellar soft-crystal by a smectic-like mesophase. As a side-effect, however, a lateral shrinking of the siloxane layers (and thus of the entire lamellae) is observed, in relation to the polydispersity of the long siloxane chains, that mimic a partial bilayer configuration. More detailed structural information can be found in the Appendix A (Appendix A).

It is worth noting that the lamellar structure observed for all siloxane-functionalized CBP molecules differs quite significantly from the unsubstituted CBP crystal structure. Actually, neat CBP molecules self-assemble into herringbone rows along which the carbazole rings stack into columns [47]. Successive herringbone rows then fit one into the other with close-packing of the carbazole and biphenyl units, as illustrated in Figure 8. The whole results clearly demonstrate the strong microsegregation ability of siloxane chains which is able to impose, not only the lamellar organization of the molecules but also the lateral packing of the CBP cores, by forcing molecular interactions through carbazole units. Siloxane chain functionalization then constitutes a powerful tool to control molecular arrangement and molecular packing. Depending on the location, number, and length of the siloxane chain, it is possible to tune the organization of the molecules and their molecular interactions.

Lastly, the spontaneous alignment observed for the siloxane-functionalized solid-state CBP derivatives should be addressed. Actually, when CBP-2Si_3_ was deposited as a thin film by spin coating, the siloxane layer planes were found to spontaneously align parallel to the substrate. This effect is most likely driven by the very low surface energy of ODMS (around 20–22 mN m^−1^) [13]. Siloxane-containing molecular systems should minimize their energy by preferably orienting the siloxane chains at the interface with air (and probably with the glass substrate also), thereby imposing the orientation of the lamellar organization parallel to the substrate on the whole film thickness [48,49]. Siloxane-functionalization then turns out to be a valuable tool for controlling the morphology of functional organic materials.

### 2.3. Photophysical Properties in Solution

The photophysical properties of the dyes were inspected in a dichloromethane solution (Appendix A). The absorption peaks observed in CBP at 340 and 293 nm are associated with transitions localized on the carbazole units while the additional absorption band at 317 nm is attributed to transitions involving the central benzidine group [35]. As stated in another study [50], the presence of both carbazole and benzidine characteristics in the absorption spectrum of CBP indicates that the electron density in the ground state is delocalized over the whole chromophore. As can be seen in Appendix A, the UV-visible absorption spectra of the different CBP functionalized with the siloxane chains are rather similar to that of CBP. This implies that the functionalization of the siloxane side chains on the carbazole units does not significantly affect the delocalization of the electron density in the ground state. The only differences appear on the redshift of the lowest absorption bands in energy when transitioning from CBP to CBP-2Si_3_ (or CBP-2Si_n_) to CBP-4Si_3_. As shown in Appendix A, a gradual red-shift of the emission spectra is also observed when transitioning from CBP to CBP-2Si_3_, CBP-2Si_n_, and CBP-4Si_3_. The fluorescence properties of CBP involve predominantly the central benzidine part of the molecule and exhibit some charge transfer character that can make it sensitive to the polarity of the environment [50]. However, previous studies have shown that the siloxane chains present a low polarity, similar to that of alkane chains [51], and we thus exclude changes in the polarity of the local environment as a reason for the different photophysical properties of the siloxane-based compounds. To gain further insights, quantum chemistry calculations were carried out to estimate the highest occupied molecular orbital (HOMO) and lowest unoccupied molecular orbital (LUMO) distributions, oscillator strengths and the first excited-state singlet energies in CBP and CBP functionalized with 2 and 4 propyl side chains (CBP-2prop and CBP-4prop, respectively) in the gas phase using time-dependent density functional theory (TD-DFT) with the B3LYP functional in the 6–31 G basis. Siloxane chains were removed for these calculations to reduce their computing time. In good consistency with a previous study devoted to CBP derivatives [52], it can be seen in Figure 9 that the HOMOs of CBP, CBP-2prop, and CBP-4prop are delocalized over the whole molecules while their LUMOs are localized mainly onto the central biphenyl. Substituting propyl side chains to the carbazole units of the CBP core is also found to hardly affect the dihedral angle at the ground state between the phenyl rings (36.4° for CBP, 36.3° for CBP-2prop and 36.2° for CBP-4prop). More noticeably, this substitution leads to some changes in HOMO/LUMO energy levels and to a gradual decrease in the singlet energy together with a slight increase in the oscillator strength (see Table 2). Overall, these calculations suggest that the small redshift of both absorption and steady-state emission spectra in Appendix A should not be attributed to a change of planarization of the molecules, but rather to the variation in their electronic properties induced by the electro-donating character of the grafted chains.

### 2.4. Photophysical and Charge Transport Properties in Thin Films

The functionalization of the CBP molecule by the siloxane hybrid side-chains has been found to strongly modify the structural properties in thin films, which, in turn, should have a significant impact on their photophysical and charge transport properties. Figure 10 displays the thin film UV/visible absorption spectra of the siloxane-functionalized CBP derivatives with their different phases, as well as of the neat CBP in its glassy amorphous and crystal states. The glassy amorphous CBP film was obtained immediately after spin-coating deposition of the CBP solution in 1 wt.% chloroform, while the crystalline state was formed after allowing the same film to stand for more than one day at room temperature. All data in solution and solid state are summarized in Table 3.

Unlike the results obtained in solution (Appendix A), the absorption and photoluminescence spectra in thin films that are displayed in Figure 10 and Figure 11 show significant differences depending on the nature of their condensed phase. The absorption spectrum of the crystalline CBP thin film exhibits a substantial redshift compared to its glassy film with a tail at longer wavelengths, which can be explained by the particular CBP molecular packing and the possible contribution of scattering by small crystallites [53]. Noticeably, the absorption spectrum of the soft crystal CBP-2Si_3_ film shows different features compared to that of crystalline CBP. In contrast to what is observed in dichloromethane solution, the relative intensity of the absorption bands in CBP-2Si_3_ film is modified with, in particular, a decrease in the absorption of the peak associated with the benzidine unit together with a hypsochromic shift. When compared to the spectra of the two other siloxane-containing CBPs, soft-crystal CBP-2Si_3_ also shows a different absorption spectrum with more intense bands at 300 and 347 nm, which are characteristics of the carbazole unit. By contrast, CBP-4Si_3_ and CBP-2Si_n_ spectra are very similar indeed and strongly resemble all CBP derivatives spectra measured in solution (Appendix A). This is presumably due to the rather weak intermolecular interactions between aromatics cores taking place in the liquid state and the fluid mesophase (Figure 7). Finally, the absorption spectrum of the non-functionalized CBP in its glassy amorphous state strongly resembles the ones of the “fluid” siloxane-functionalized CBP derivatives CBP-4Si_3_ and CBP-2Si_n_.

Figure 11 displays the steady-state photoluminescence spectra of the CBP derivatives in thin films. Compared to the CBP glassy film, the emission spectrum of the CBP crystal exhibits a substantial redshift together with a well-resolved vibronic structure exhibiting two vibronic peaks and one shoulder in the range between 370 and 410 nm. Noticeably, the fluorescence spectra of both CBP-2Si_n_ and CBP-4Si_3_ show similar features (see Table 3) and the slight red-shift of the emission of CBP-4Si_3_ is presumably due to the effects of the two additional electron-donating side chains in CBP-4Si_3_ as compared to CBP-2Si_n_ on the electronic properties, similarly to what is observed in solution. The most intriguing result in Figure 11 comes from the emission spectrum of CBP-2Si_3_. While its emission spectrum shows the same vibronic structure as the other siloxane-containing CBP derivatives, the emission of CBP-2Si_3_ is blue-shifted by more than 20 nm. Carbazole derivatives are known for their tendency to form excimers in which interacting carbazole units are stacked in an overlapping sandwich-like configuration [54]. However, the fluorescence of CBP is dominated by the properties of the central biphenyl part of the molecule and it has been shown that this compound does not exhibit excimer emission in thin films [52]. In addition, the fluorescence of highly twisted CBP derivatives was found to be dominated by the individual properties of the *N*-phenylcarbazole units and to show a significantly blue-shifted emission as compared with the spectrum of CBP films [52]. In this context, the most plausible explanation for the blue-shift of the emission is that CBP-2Si_3_ molecules adopt in the condensed phase a more twisted geometry with a larger torsion angle between the two phenyl rings of the benzidine core. This would also be consistent with the observed decrease in the absorption band of the benzidine moiety in the absorption spectrum of CBP-2Si_3_ film and with the fact that the thin film shows a blue shifted emission as compared to the solution [55]. It should also be emphasized that the X-ray scattering results indicated that the structure and morphology of the CBP-2Si_3_ film is lamellar with layers oriented in the direction parallel to the substrate. The interactions within the layers of aggregated CBP rings affect the emission with respect to isolated molecules in solution [56]. Additionally, there might also be an impact of molecular architecture and packing on the dihedral angle between the two *N*-phenylcarbazole units. To gain further insights, we looked at the molecular geometry at the DFT level in order to examine the potential energy landscape of the CBP-2Si_3_ molecule (Appendix A). The most stable molecular geometry is obtained for an angle around 30°, which is in good consistency with a previous report devoted to CBP [52]. However, the calculations show that there is another energetically favorable minimum for an angle around 50° and it is, therefore, possible that the average dihedral angle in CBP-2Si_3_ films deviates from the average value in solution, which could explain a part of the frequency shift.

The photoluminescence quantum yield (*PLQY*) of the CBP derivatives was then measured in thin films. As displayed in Table 3, the siloxane-functionalized CBP series show a substantial decrease in PLQY as the degree of order increases, from 0.58 (Liquid), 0.35 (Liquid crystal) to 0.16 (Soft crystal). However, soft crystal CBP-2Si_3_ gives a lower *PLQY* than classical crystal CBP (0.36), which means that molecular organizations and conformations in both systems have different efficiencies upon luminescence quenching. If we compare the molecular organization in CBP and CBP-2Si_3_, which are schematically represented in Figure 7 and Figure 8, the herringbone organization in the CBP crystal might involve a lower aromatic core overlap and thus a weaker quenching of the emission due to intermolecular interactions. In addition, the twisting of the benzidine core suggested by the absorption and steady-state fluorescence spectra of CBP-2Si_3_ would presumably lead to a reduction in the oscillator strength and a substantially lower *PLQY* value. Regarding the liquid crystalline CBP-2Si_n_, the CBP cores in this system self-arrange in a similar way as CBP-2Si_3_ but the efficiency of the benzidine core interactions and ultimately the quenching is altered by the less regular smectic in-plane order. Finally, these interactions are further reduced in the liquid phase, leading to the highest *PLQY* for CBP-4Si_3_. The luminescence of this liquid material even overcomes that of glassy amorphous CBP, for which *PLQY* is also enhanced by structural disorder, relative to crystalline CBP.

The CBP molecule has been intensively used as a host material in organic light-emitting diodes [57] and its neat film shows hole and electron mobilities on the order of 10^−3^–10^−4^ cm^2^ V^−1^ s^−1^ [58,59]. A previous work has characterized the charge carrier mobilities of siloxane-containing oligofluorene derivatives using the time-of-flight technique (ToF). The electron and hole mobilities in liquid oligofluorene were found to be on the order of 10^−4^ cm^2^ V^−1^ s^−1^ and comparable with the values measured in solid thin films of other fluorene derivatives [27]. In this context, it was relevant to characterize the charge transport properties of the siloxane containing CBPs but, in contrast with what was observed in liquid fluorene derivatives, their investigation by ToF turned out to be unsuccessful (see Appendix A). The measurements carried out on commercial ITO-covered liquid crystal cells filled by capillarity with materials in their liquid state only led to a poor response indicating a low charge carrier mobility with estimated values well below 10^−6^ cm^2^ V^−1^ s^−1^ for all materials. Regarding the soft crystalline CBP-2Si_3_, the film shows no domain orientation when implemented as the semiconducting layer of the measuring device. Charge transport of CBP-2Si_3_ was therefore jeopardized by the insulating siloxane layers interrupting the conduction pathways towards electrodes. In the case of CBP-4Si_3_, it can be assumed that the significantly reduced intermolecular interactions between CBP units, as confirmed by its high *PLQY* value, lead to poor conductive pathways. The same effect presumably occurs for CBP-2Si_n_ in conjugation with the dilution of the low-efficient conduction pathways in the high-volume fraction of insulating siloxane.

The results obtained in this study indicate that these CBP derivatives exhibit promising photophysical properties for organic optoelectronics, but their potential use is still strongly limited by their charge transport properties. One aspect potentially detrimental to charge transport in the liquid and liquid crystal states of these systems is the location of the siloxane chains onto the carbazole end units. By considering the molecular organization as depicted in Figure 7, it is possible that the insertion of voluminous side chains directly to the carbazole end units alter the efficiency of conduction pathways by reducing the *π*-orbital overlap between carbazole units. Consequently, there is still a possibility to improve the molecular design and get liquid or liquid crystalline CBPs with enhanced charge transport performances.

## 3. Conclusions

Siloxane substituents can be seen as an alternative to alkyl chains for the control of the molecular organization in organic thin films. Thus, by functionalizing *π*-conjugated molecules with siloxane chains, mesomorphic organizations are readily obtained by innate segregation between the siloxane chains and the conjugated units. At the same time, crystallization is drastically hindered, usually in favor of a glass transition at a very low temperature, leading to the formation of soft and fluid material in an ambient environment. In a previous work, we demonstrated that CBP, a conjugated molecule of interest for organic optoelectronics and well-known in the area of OLEDs, could become liquid at room temperature through the introduction of siloxane-terminated side-chains, whereas the unsubstituted material is crystalline and melts at 270 °C. In the present study, the design of the siloxane side-chains allowed to vary the self-organization of the materials between a three-dimensional lamellar soft-crystal with two short siloxane chains, a nanosegregated liquid freezing only at −62 °C with four chains, and a room-temperature smectic liquid-crystal with two longer chains. The photophysical properties of the films were then investigated in the different materials and could be correlated to their molecular organizations. In particular, the liquid CBP functionalized with four siloxane chains was found to exhibit a *PLQY* of 58%, higher than that of glassy CBP, due to a reduction in the intermolecular interactions between neighboring conjugated cores. However, the results also indicate that those siloxane-containing CBP derivatives exhibit poor charge transport properties, which seriously limit for now their potential use in organic optoelectronics. While the outcome of this work is highly relevant for rationalizing the role of siloxane chain functionalization in the control of the molecular organization, further efforts are still required in terms of molecular design to obtain high-performance functional siloxane containing optoelectronic materials based on a CBP core.

## Data Availability

Data is available in this article and its Appendix A.

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
