# Peer review of "Control of the Organization of 4,4′-bis(carbazole)-1,1′-biphenyl (CBP) Molecular Materials through Siloxane Functionalization"

_molecules, 2023, doi:10.3390/molecules28052038_

Round 1
Reviewer 1 Report
The manuscript presents the functionalization of 4,4′-bis(carbazole)-1,1′-biphenyl (CBP) with siloxane units of various length. It leads to significant destabilization of the crystal phase, which enables obtaining the less-ordered states in lower temperatures. The phase identifications is made based on the results of polarizing optical microscopy and X-ray diffraction. The photophysical properties in solutions and of thin films are investigated. I recommend the publication of this work after minor revision. My remarks are given below:
lines 58, 65, 66, 67, 73, 464 – there are some spaces and hyphens “ -” in the text which probably should be absent
line 176 – Fig. 4d shows the supercooled liquid phase, not mesophase, according to the main text (line 222)
Author Response
We are thankful to the reviewer for pointing out miswritten characters and errors in the text.
1) lines 58, 65, 66, 67, 73, 464 – there are some spaces and hyphens “ -” in the text which probably should be absent
We went through the whole manuscript to correct all typo errors, including the one mentioned.
2) line 176 – Fig. 4d shows the supercooled liquid phase, not mesophase, according to the main text (line 222)
The reviewer is perfectly right, and we are thankful for noticing a mistake in the figure caption of Fig. 4. It was therefore corrected as mentioned below:
Old caption of figure 4 in lines 174-176: "SWAXS patterns of the CBP materials recorded at room temperature: a) CBP-2Si3 (solid state), b) CBP-4Si3 (liquid phase), c) CBP-2Si10 (pristine, solid state) and d) CBP-2Si10 (after melting, in the mesophase)."
New caption of figure 4: "SWAXS patterns of the CBP materials: a) CBP-2Si3 (solid state at 20°C), b) CBP-4Si3 (liquid phase at 20°C), c) CBP-2Si10 (mesophase at 10°C) and d) CBP-2Si10 (supercooled liquid phase at 20°C)."
Reviewer 2 Report
This article is about Control of the organization of 4,4′-bis(carbazole)-1,1′-biphenyl 2 (CBP) molecular materials through siloxane functionalization. It is very well written with recent important references. However, prior to publish some minor revision should be taken into account.
1. Explain the reason for dual peak in the emission curve.
2. How the authors have calculated the quantum yield? And how it relates with intensity photoluminescence peak?
3 Have authors also calculated the full width and half maxima (FWHM)? If yes, kindly mention.
4. Kindly add some more recent references related to your work in the introduction section.
Author Response
We are thankful to the reviewer for his careful reading and for his valuable suggestions. We answered all the points he/she addressed as detailed in the joint document.

Reviewer 3 Report
Reviewer’s comment
This original article describes the self-assembling behaviors of four CBP compounds bearing oligosiloxane
unit at the extremity of sidechain. In this study. the authors investigated their phase transition behaviors by
polarizing optical microscopy, differential scanning calorimetry and X-ray diffraction measurements.
Furthermore, photophysical properties of oligosiloxane-modified CBPs were experimentally and theoretically
revealed by UV-vis and photoluminescent spectroscopies as well as DFT calculation. The concept described in
this article will be helpful to understand the self-organization of oligosiloxane-modified aromatic compounds as
well as the precisely control of nanostructure based on “nano-segregation”. Although, the present manuscript has
some issues as described later, I think that the overall is good. Therefore, I recommend that the manuscript should
be acceptable after minor revision on the findings that I point out as following.
Comments on the manuscript:
1. (p.3, Figure 1)
The chemical structure of CBP-2Si10 in Figure 1 should be changed. I think that the description of “n~8” is
unsuitable. The subscript in the chemical structure must be “n”. The description of “n ≈ 8” should be placed in
the side of “CBP-2Si10”.
2. (p.3, Synthesis; Characterization of CBP-2Si10 in ESI)
Because the oligomer uniformity affects to the physical properties, the polydispersity index of CBP-2Si10
should be estimated by size exclusion chromatography (SEC). Please show the value of polydispersity index
and provide SEC trace. In addition, please mention the assignment of following signals on 1H- and 13C-NMR
spectra of CBP-2Si10. The corresponding signal are appeared at = 1.3, 3.5 ppm and C = 30 ppm. Please
justify the chemical purity.
3. (p.5, Figure 2)
Why is the indicated temperature range of the DSC curves not same? Please explain.
Are the DSC curves shown in Figure 2 for the first cooling and second heating scans? If so, the authors should
state in plot or caption.
4. (p.5, POM study; p.8, lines 231-232)
Please display the POM images in the crystal (soft-crystal) phase of CBP-2Si3 at room temperature. In p.8,
why did the authors use the “soft crystal phase” not “crystal phase” for the solid state of CBP-2Si3 at room
temperature. Please briefly mention in the manuscript.
5. (p.6, Figure 4)
Although the shoulder found at q = 0.16 Å-1 in Figure 4c is possible to be considered as the diffraction of a
liquid-crystalline columnar structure such as (110) plane, is the attribution of the liquid crystalline phase of
SmE appropriate? Because fan-shaped textures can be also observed in some liquid-crystalline columnar
phases, it cannot let go of the possibility that molecules of CBP-2Si10 form a complicated columnar structure
such as B1 structure in the room temperature mesophase. Please briefly explain. If possible, I recommend to
the remeasurement of XRD for the pristine sample of CBP-2Si10. I think the comparison between the XRD
results and theoretical molecular length for all those compounds by MM2 or DFT calculation is preferable.
6. (p.9, Figure 6; p.10, Figure 7, 8; Figure S6 in the ESI)
I firstly imagined the self-assembled structures of bent-core liquid crystals (i.e. B1, B1rev, B2 etc. [Top. Curr.
Chem. 2012, 318, 281-330; J. Am. Chem. Soc. 2000, 122, 1593.]) from Figure 6, because the molecular shapes
of CBP-2Si3 and CBP-2Si10 should be not straight bar but bended. I hope the authors discussion in this article
about the crystalline and liquid-crystalline structures of CBPs from the viewpoint of the similarity in molecular
shapes between the CBPs and bent-core molecules bearing oligosiloxane units [J. Am. Chem. Soc. 2004, 126,
14312; J. Am. Chem. Soc. 2006, 128, 3051; Chem. Mater. 2015, 27, 4525; CrystEngComm 2020, 22, 8412.].
For Figures 6, and 7, I request the replacement of illustrations to the other one, which is more visually
imaginable of the molecular orientation of CBPs in the crystal or liquid-crystal structures. I think the present
illustrations are unsuitable, because the oligosiloxane units are linked by covalent bond via aliphatic spacer to
central CBP core. In addition, I do not deny the existence of dynamic short-range molecular order in a liquid
phase. However, I think the illustration of the “liquid” in Figure 7 and Figure S6 is inapt as following reasons.
The intermolecular distance in isotropic liquid phase should be longer than that in mesophases. In addition,
the macroscopic layer structure and molecular order must be broken in the liquid phase so that optically
isotropic. Thus, the illustration should be fixed as considering the above issues. Since these two figures are
overlapped, the authors should delete either one. If the authors can use Chem3D or similar modelling software,
the three-dimensional molecular models should be constructed by using the software. The three-dimensional
molecular models must be useful for schematically depicting of molecular packing in several phases. Although
I guess that Figure 8 could be deleted depending on the situation, the change of illustrations in Figure 8 should
be also required.
7. (p.12-15, photophysical property)
Please discuss the difference in the optical band gaps and emission maxima between theoretical values and
experimentally obtained values.
8. (p.15, TOF mobility)
I think that the measurement conditions of the TOF measurement such as excitation light source, sample
thickness and applied voltage so important. Please described the conditions. If the authors want to measure the
TOF mobility in crystalline phase, the thick sample is favorable. However, the thick sample often provides the
dispersive photocurrent decay. In order to obtain the large domains with poor trapping sites, the cooling rate
must be precisely controlled. In addition, the additional purification can be helpful, because the ionic impurities
inhibit efficient carrier transport. If the authors are possible to carry out it, I recommend the remeasurement
according to the above cautions. If CBP-2Si10 forms B1-like structure at room temperature, I guess that the
electron carrier transport should be hindered so that low TOF photocurrent. In contrast, CBP-2Si10 forms good
SmE structure, high TOF photocurrent should be found. Therefore, I am interested in their carrier transport
properties.
9. (In the whole of the manuscript and ESI)
Please check the font of physical quantity. The representative alphabet of physical quantity must be italic
font. Furthermore, please check the garbled characters written in symbol font. The unit notation rule should be
unified. The unit of “cm2/V.s” must be changed to “cm2 V−1 s−1”. In case of SI derived unit, the half-space
should be inserted not “.” between each SI units.
Summary:
The concept described in this article will attracts broad research interests for material scientists as well as the
expert of liquid crystals. The authors have been evaluated four-types of CBPs and got good results, the discussion
described in present manuscript seems to be reasonable except for the issues that I point out above. I recommend
that the manuscript acceptable after minor revision.

Author Response
We are grateful to the reviewer for his careful reading and his valuable comments. All comments have been succesfully addressed, as detailed in the document here enclosed.

Reviewer 4 Report
This manuscript can no doubt be published in ‘Molecules’ with little or no editing.
List of minor сorrections is contained in the attached pdf file

Author Response
We are grateful to the reviewer for his careful reading and his valuable comments. All comments have been succesfully addressed as detailed in our response here enclosed.
